# Automated Feeding Behaviors Associated with Subclinical Respiratory Disease in Preweaned Dairy Calves

**DOI:** 10.3390/ani10060988

**Published:** 2020-06-05

**Authors:** Catie Cramer, Kathryn Proudfoot, Theresa Ollivett

**Affiliations:** 1Department of Animal Sciences, Colorado State University, Fort Collins, CO 80521, USA; 2Atlantic Veterinary College, University of Prince Edward Island, Charlottetown, PE C1A 4P3, Canada; kproudfoot@upei.ca; 3Department of Medical Sciences, School of Veterinary Medicine, University of Wisconsin-Madison, Madison, WI 53706, USA; ollivett@wisc.edu

**Keywords:** automated calf feeder, bovine, calf lung ultrasound, pneumonia

## Abstract

**Simple Summary:**

Pneumonia in dairy calves impacts animal welfare and is costly to producers. Feeding behavior might be a useful aide for early disease detection before the clinical signs of disease are apparent. This study compared feeding behavior among calves with either pneumonia and no outward signs of respiratory disease, pneumonia plus outward signs of disease, and calves without any respiratory disease. Although unaffected calves and calves with inapparent pneumonia had similar feeding behaviors, we found that calves with pneumonia plus visible signs of respiratory disease drank slower than the others. Therefore, feeding behavior may not be useful to detect calves with pneumonia before clinical signs are apparent.

**Abstract:**

Little is known about feeding behaviors in young dairy calves with subclinical respiratory disease (SBRD). The objective of this study was to determine if calves with their first case of SBRD exhibit different feeding behaviors during the 7 d around detection, compared to calves with their first case of clinical BRD (CBRD) or without BRD (NOBRD). Preweaned, group-housed dairy calves (n = 103; 21 ± 6 d of age) underwent twice weekly health exams (lung ultrasound and clinical respiratory score; CRS); health exams were used to classify the BRD status for each calf: SBRD (no clinical signs and lung consolidation ≥ 1cm^2^; n = 73), CBRD (clinical signs and lung consolidation ≥ 1cm^2^; n = 18), or NOBRD (never had lung consolidation ≥ 1cm^2^ or CRS+; n = 12). Feeding behavior data (drinking speed, number of visits, and intake volume) were collected automatically. Calves with SBRD and calves with NOBRD had similar drinking speeds (782 vs. 844 mL/min). Calves with CBRD drank slower than both calves with SBRD (688 vs. 782 mL/min) and NOBRD (688 vs. 844 mL/min). There was no effect of BRD status on any other behavior. Feeding behavior was not an effective means of identifying calves with SBRD.

## 1. Introduction

Based on producer reporting, clinical bovine respiratory disease (BRD) affects 12% of preweaned calves in the United States and accounts for 22% of all preweaned calf deaths [1]. Early disease detection and intervention likely ameliorates the negative consequences of BRD [2]. However, producers and veterinarians struggle to identify all calves with BRD [3,4,5]. Detecting sick calves in group housing is particularly challenging, due to the difficulty of observing individual animals and our limited knowledge of how sick calves behave in a social environment. These challenges are compounded by the prevalence of subclinical BRD (lung consolidation, but no visible signs of disease) among preweaned dairy calves, which can range from 23% to 69% [6,7].

Automated calf feeders are increasing in popularity on dairies and offer an additional way to detect sick calves with more data and potentially less calf handling than traditional screening methods. Feeding behavior data collected from automated feeders have shown promise to identify sick calves identified using clinical detection tools. Borderas et al. [8] observed that calves fed ad libitum milk with gastrointestinal illness, respiratory illness, or a combination thereof, drank 2.6 L/d less and had 2.4 fewer visits per day compared to calves without illness. A recent study observed that sick calves (defined as diarrhea, respiratory disease, or ill thrift) drank 183 mL/min slower, drank 1.2 L/d less, and had 3.1 fewer unrewarded visits (visits to the feeder without milk) compared to healthy calves [9]. Previous work indicates that feeding behaviors change around the time of illness, but disease was defined broadly and used clinical (visible signs of disease) detection tools [8,9].

Clinical detection tools include visual observations and lung auscultation [8,10], or the Wisconsin Calf Respiratory Scoring chart [9,11] to define BRD. Lung auscultation and BRD scoring systems that rely solely on clinical signs lack sensitivity [12] and require handling individual animals. The limitations of these clinical BRD detection methods preclude our ability to accurately identify all diseased calves. Therefore, a large knowledge gap exists because we cannot fully understand the behavioral changes associated with this complicated syndrome due to the potential misclassification of calves with BRD.

Calf lung ultrasound has been validated in dairy calves as a rapid, on-farm diagnostic tool for the identification of lung consolidation associated with BRD [13,14]. Lung ultrasonography can accurately detect lung consolidation, regardless of the calf’s clinical status [14]. The sensitivity and specificity of ultrasound are high (94% and 100%, respectively) for detecting the lung lesions associated with BRD in the absence of clinical signs [14]. These advantages plus the commonness of subclinical respiratory disease (SBRD) provides an impetus for using lung ultrasound to study the relationships between feeding behavior and BRD. Ideally, highly sensitive BRD detection methods, such as lung ultrasound, would be used to validate on-farm automated BRD detection tools. Therefore, the objective of this study was to determine if calves with subclinical BRD (lung consolidation, but no outward signs of disease) exhibit different feeding behaviors compared to calves with clinical BRD (lung consolidation with outward signs of disease) or calves without BRD during the 3 d before, the day of, and the 3 d after BRD detection. We hypothesized that calves with subclinical BRD would have deviations in feeding behaviors compared to calves without BRD, but to a lesser degree than calves with clinical BRD.

## 2. Materials and Methods

Data collection for this study took place between February and August 2016, on a commercial dairy cattle facility in Ohio, USA. The Institutional Animal Care and Use Committee at the University of Wisconsin-Madison (A005049-A03) approved this study. Calf management prior to study enrollment is previously described in detail [7].

Calves were enrolled into the study upon entry to the automated calf feeder barn at 21 ± 6 d of age. Calves were recruited for enrollment from February to July 2016, and were eligible for study enrollment if they entered the automated calf feeder barn by 30 d of age. Calves were followed until 50 d of age. Four automated calf feeders (Lely Calm calf feeder, Lely North America, Pella, IA, USA) served 2 pens each, with 1 nipple per pen. Calves remained in their original pen until weaning. Starter grain was available ad libitum beginning at 3 d of age. Calves had ad libitum access to whole milk from early lactation cows until 40 d of age; from 41 to 46 d of age, calves were limited to 8 L of milk/d and then ramped up from 8 to 9.7 L/d over the course of 6 days. From 47 to 56 d of age, calves were stepped down from 9.7 to 9 L/d. The age of the calf was entered into the automated feeding program, to ensure calves were on the correct feeding stage based on their age. The meal size was 1 L/meal until 40 d of age, 1.5 L/meal from 41 to 46 d of age, and 2 L/meal from 47 to 56 d of age. Medicated milk replacer (32 g milk replacer per 1 L whole milk) was supplemented in the calf feeders (Renaissance Nutrition Inc., Roaring Spring, PA, USA).

The following feeding behaviors were automatically generated by the milk feeder: average daily drinking speed (mL/min), total daily milk intake (L/d), number of unrewarded visits per day (number of visits to the feeder without milk), and number of rewarded visits per day (number of visits to the feeder with milk). Average meal size consumed by the calf across one day was calculated by the following: daily milk intake/rewarded visits. Research staff downloaded the feeder data weekly to Microsoft Excel 2010 (Microsoft, Redmond, WA, USA).

Research staff performed health examinations on all enrolled calves twice weekly between 09:00 and 13:00 h until 50 d of age [7]. Three research staff members performed a clinical respiratory score [11]; one experienced researcher (CC) trained two research staff members on how to properly score calves prior to the study. Health examinations were recorded using the Wisconsin Calf Health Scorer App (https://www.vetmed.wisc.edu/dms/fapm/apps/chs.htm), and included a clinical respiratory score (CRS; [7,11]), a 6 level ultrasound score [6], and a fecal score (0 = normal fecal consistency; 1 = semi-formed, pasty; 2 = loose but stays on top of bedding; 3 = watery, sifts though bedding; [15]). Briefly, the CRS assigned 0 (normal) to 3 (severely abnormal) points for each of the following categories: nasal discharge, eye discharge, ear position, cough, and rectal temperature. Calves were considered positive for clinical respiratory disease (CRS+) when 2 categories with a score of 2 or greater were observed [11]. The numerical rectal temperature was also recorded and fever was defined as ≥39 °C. Diarrhea status was recorded and was defined as fecal score ≥ 2 [15]. Ultrasound examinations of the left and right lungs were scanned in a systematic manner, using a technique and scoring system previously described in dairy calves [6,7,14]. One researcher (CC) performed all lung ultrasound exams. The ultrasound scoring system ranged from 0 to 5 (0 = normal or ≤1cm^2^ consolidation; 1 = diffuse comet tails; 2 = lobular pneumonia: consolidation ≥1 cm^2^; 3 = lobar pneumonia, 1 entire lung lobe consolidated; 4 = lobar pneumonia, 2 entire lung lobes consolidated; 5 = lobar pneumonia, ≥3 entire lung lobes consolidated). Ultrasound scores ≥2 indicated that lung consolidation ≥1cm^2^ was present. Lung characteristics such as necrosis, abscessation, or pleural effusion were also noted.

Bovine respiratory disease was defined by the presence of lung consolidation, with or without the presence of clinical signs (i.e., cases that producers could identify using visual signs), resulting in three categories of BRD status based on a calf’s first BRD event: (1) clinical BRD (CBRD): CRS+ with lung consolidation ≥1cm^2^ at their first detected BRD event, (2) subclinical BRD (SBRD): lung consolidation ≥1cm^2^ and not CRS+ at their first detected BRD event, or (3) without BRD (NOBRD): never CRS+ and never had lung consolidation ≥1cm^2^ throughout the study [16]. All CBRD calves were treated with an antibiotic, according to a standardized treatment protocol developed in cooperation with the farm management and farm veterinarians [7]. Calves identified with SBRD at their first BRD event were not treated with an antibiotic; these calves may have later received an antibiotic if they were subsequently identified as CRS+.

Feeding behavior data were summarized for the 3 d before (d − 3, d − 2, d − 1), the day of (d 0), and 3 d (d + 1, d + 2, d + 3) after BRD identification for CBRD and SBRD calves. To control for age as a potential confounder, feeding behaviors from NOBRD calves were selected from similar age ranges to SBRD and CBRD calves. The day selected for feeding behavior analysis for NOBRD calves was set using the mean age of all calves with SBRD and CBRD (31 ± 6 d old). A one-way ANOVA was used to ensure that age was not different between BRD categories. Calves with missing feeder data (due to computer, electrical, or internet failure) were not included in the final analysis. Data were stored, cleaned, and analyzed using Microsoft Excel (Microsoft, Redmond, WA, USA) and SAS (version 9.4; SAS Institute Inc., Cary, NC, USA). Sample size calculations performed a priori determined the necessary sample size to be 35 calves per group using a power of 0.8 and a mean ± SD difference of 3 ± 4.3 L of milk intake per day between sick and healthy calves [8].

The experimental unit was the calf for all analyses. Calves in each BRD status were mutually exclusive and each calf only had one BRD status for analyses. Calves identified with diarrhea from d − 3 to d + 3 relative to BRD identification were removed from analysis. Data were screened for outliers using visual assessment and for normality using PROC UNIVARIATE in SAS. Continuous variables were assessed for normality using the Shapiro Wilk test and visual inspection of histograms. The outcomes of interest were daily average drinking speed (mL/min), milk intake (L/d), average meal size (L/visit), number of unrewarded visits (no./d), and number of rewarded visits (no./d) [8,9]; each outcome was modeled separately. The predictor of interest was BRD status (CBRD, SBRD, or NOBRD).

Proportions of calves by sex, breed, and fever (rectal temperature 39 °C on the day for which feeding behavior was analyzed) for each BRD status were compared using chi-squared or Fisher’s exact tests (Table 1). Explanatory variables were screened for inclusion in full multivariable models using univariable analyses, which are previously described in detail [7]. Briefly, explanatory variables were tested for an association with each outcome of interest and explanatory variables with *p* < 0.2 were initially offered to the full multivariable models [17]. Collinearity between CRS scores and rectal temperature scores was assessed using Spearman correlation coefficients [18]. Rectal temperature scores and CRS scores were highly correlated (correlation coefficient = 0.7; *p* < 0.001), and as such, only BRD status was included in analyses. Three separate multivariable linear mixed models (MIXED procedure in SAS) were used to determine if BRD status (predictor) was associated with the following outcomes: drinking speed, milk intake, and average meal size. The GLIMMIX procedure with Poisson distributions (specified using dist = function) was used to determine whether BRD status (predictor) was associated with number of unrewarded and rewarded visits to the feeder (outcomes), using two separate models [19].

All full multivariable models included BRD status, day (d − 3 to d + 3), sex (male = 0; female = 1) and breed (Holstein = 0; Jersey = 1) as fixed effects. All models also included day as a repeated measure and calf as a random effect. Forward selection followed by backwards stepwise elimination was used to select only variables that were significant at alpha ≤0.05 level for the final models. A variable was considered a potential confounder if it met previously described criteria [17]; to detect confounding prior to the final elimination of a variable, a change in estimate criterion of ≥20% for the predictor of interest was used. All potential interactions with BRD status and the remaining variables in the model were assessed, and interactions with *p* > 0.05 were removed. *p*-values in multivariable models were adjusted for multiple comparisons using Tukey’s method for multiple comparisons. Final models included only BRD status and breed.

Predicted means for feeding behaviors were assessed using the LSMEANS statement and differences between BRD statuses were assessed using differences in LSMEANS. Type 3 tests of fixed effects were used to determine significance at the alpha ≤0.05 level. Residuals were plotted and used to check model assumptions. Measures of central tendency for raw data are presented as mean ± SD or median (1st quartile, 3rd quartile). Estimates from multivariable models are presented as mean ± SE.

## 3. Results

Detailed calf numbers from birth to study enrollment were previously described [7]. One calf with a Cook’s Distance >4/n was removed as an outlier. This calf had a small birth weight, poor growth, and no signs of respiratory, diarrhea, navel, or joint conditions. Sixteen calves were identified as CRS+, but did not have lung consolidation and were therefore not included in analysis. Calves identified with diarrhea from day − 3 to d +3 were not included in analysis (n = 5 from CBRD and n = 8 from SBRD). Calves included in final analysis (n = 103) were 21 ± 6 d (mean ± SD) at entry to the automated calf feeder barn and were housed in groups of 13 ± 3. Calf descriptive information and the proportion of calves with fever are presented in Table 1. Seventeen percent (18/103) and 71% (73/103), of calves were identified with CBRD and SBRD at their first BRD event, respectively. Twelve percent (12/103) of calves were never identified with CBRD or SBRD and were therefore categorized as NOBRD. The age (mean ± SD) of calves with NOBRD (33 ± 4 d) on the day selected for feeding behavior analysis was similar to the mean age of both calves with SBRD (32 ± 6 d) and the mean age of calves with CBRD (31 ± 6 d), on the day of BRD detection (*p* = 0.57). The proportion of calves identified with each BRD status was not different between pens (*p* = 0.51).

There was no difference in the amount of milk offered (*p* = 0.96) or the meal size (*p* = 0.95) due to the feeding plan between BRD statuses. Inter-observer agreement for CRS status (CRS+ versus CRS-) was determined prior to the study (kappa = 0.6) and halfway through the study (kappa = 0.7). Raw values for feeding behaviors are presented in Table 2. Day was not associated with any feeding behavior (*p* > 0.15), nor was the interaction between day and BRD status (*p* > 0.37).

Calves with CBRD drank slower than both calves with SBRD (687 ± 42 vs. 782 ± 25 mL/min; *p* = 0.02) and calves with NOBRD (687 ± 42 vs. 844 ± 51 mL/ min; *p* = 0.01; Table 3). There was no difference in drinking speed between calves with SBRD and calves with NOBRD (782 ± 25 vs. 844 ± 51 mL/min; *p* = 0.26). There was no effect of BRD status on milk intake (*p* = 0.64), average meal size (*p* = 0.79), rewarded visits (*p* = 0.26), or unrewarded visits (*p* = 0.19; model results not shown).

## 4. Discussion

To our knowledge, this study was the first to investigate if feeding behavior in calves with subclinical BRD differed from calves with clinical BRD or calves without BRD. Total daily milk intake, average meal size, number of rewarded visits, and the number of unrewarded visits did not differ between the three BRD categories. Drinking speed was the only feeding behavior that differed among the three categories of BRD status, whereby calves with clinical BRD drank slower than both calves with subclinical BRD and calves without BRD. Similarly, Knauer et al. [9] reported that calves with clinical BRD drank slower than calves without clinical BRD on the day of illness detection. The drinking speeds for sick calves reported by Knauer et al. [9] were similar to our results for calves with clinical BRD, and healthy calves in the Knauer et al. study drank at similar speeds to calves without BRD and calves with subclinical BRD in the present study.

The observation in the present study that calves with subclinical BRD drank faster than calves with clinical BRD, yet similar to calves without BRD, over the 7 d period, might be due to the increased lethargy associated with clinical BRD [5,16]. It is not uncommon to see dull or depressed calves lingering at the feeder without drinking. These lethargic calves might not have the strength or desire to drink at faster speeds, as a consequence of sickness behavior [20]. Sickness behaviors are thought to conserve energy for the febrile response to illness, thus increasing chances of survival [20]. It is possible that calves with subclinical BRD did not exhibit sickness behaviors, such as slower drinking speeds, because a smaller proportion of calves with subclinical BRD had a fever, compared to calves with clinical BRD. Although there is limited research, factors such as amount of milk fed, farm management factors, and nipple size may play a role in drinking speed variation. Based on our results, drinking speed may be useful to identify calves with clinical BRD, but this is likely not a suitable measure to identify calves with subclinical BRD.

Regarding milk intake, we did not observe a difference among BRD statuses. Similarly, Knauer et al. [9] did not observe a difference in milk intake between calves with clinical BRD and healthy calves. However, calves in Knauer et al. [9] were limit-fed on many farms and the intakes for all calves were lower compared to calves in the present study. In contrast, Borderas et al. [8] observed a trend for ill calves fed a high milk allowance to reduce their milk intake 2 d prior and the d of illness detection. However, Borderas et al. [8] combined gastrointestinal and respiratory illness into one overall category and compared ill calves to healthy, making it difficult to compare with the present study. Our results suggest that milk intake is not a useful measure to identify calves with clinical BRD, nor subclinical BRD. Since alarms on automated calf feeders are most often designed to detect reductions in milk intakes, it is unlikely that a feeder alarm would identify calves with either clinical or subclinical BRD in calf systems managed similarly to the system in the present study.

We did not observe an effect of BRD status on rewarded visits, similar to previous studies [9,21]. In contrast, Knauer et al. [9] found that calves with clinical BRD had fewer unrewarded visits on the d of treatment for BRD compared to healthy calves, whereas we observed no effect of BRD status on unrewarded visits. Calves in the present study had numerically fewer unrewarded visits compared to Knauer et al. (mean unrewarded visits: 7.5). High milk allowance has been shown to reduce the number of unrewarded visits [22]. Calves in the present study were offered ad libitum milk access until 40 d of age, whereas Knauer et al. [9] reported an average full feeding level of 9.4 L/d (range = 7–16). Therefore, varied milk allowances between studies might contribute to different findings between the relationship between BRD and the number of rewarded and unrewarded visits. Results from the present study suggest that rewarded and unrewarded visits are not ideal to identify calves with clinical or subclinical BRD when calves are fed ad libitum milk.

The interpretation of results from the present study should consider the potential biases of our study design. We based BRD status on a calf’s first BRD event after study enrollment. However, we did not enroll calves until 21 d of age, but clinical BRD [23] and lung consolidation [24] have been reported in calves younger than 3 weeks and at 7 d of age, respectively. Therefore, it is possible that calves we identified as normal were actually recovered from a BRD event prior to study enrollment. It is unlikely that changes in feeding behaviors from a BRD event prior to study enrollment would last through the study period, as Knauer et al. [9] did not observe differences in feeding behaviors between calves with clinical BRD and controls after the d of BRD identification. However, we may have introduced selection bias, as only the calves that survived until 21 d of age were eligible for study enrollment. The exclusion of calves that died prior to 21 d of age may have biased our results towards the null, as calves that died were likely most affected by disease. Age may affect how calves express sickness behavior, therefore the exclusion of feeding behavior from calves <21 d in general may also have biased our results towards the null. Young calves are less explorative than older calves [25], and may not cope as well with illness compared to older calves. A previous study observed an interaction between age and treatment (bacterial endotoxin versus control) for time spent inactive in dairy calves [26], thus, age can affect the expression of sickness behavior. Additionally, we were unable to measure starter grain intake and therefore were unable to determine if starter grain intake differed between BRD categories.

The inaccuracy of the BRD scoring systems used in the present study may have resulted in the misclassification of BRD, leading to biased estimates for feeding behaviors. When using lung ultrasound, misclassification can occur if consolidation does not extend to the lung surface, resulting in false negatives [27]. In rare cases, calves with non-aerated lung not due to BRD might be misclassified as positive [27]. Because lung ultrasound is imperfect, a recent study used Bayesian estimates to measure the sensitivity of CRS and lung ultrasound [4]. Using a cutoff of a score of ≥5, the CRS had a sensitivity of 62% (95% CI: 48–76) and a specificity of 74% (95% CI: 65–83). Using a cutoff of ≥1 cm, lung ultrasound had a sensitivity of 80% (95% CI: 66–91) and a specificity of 95% (95%CI: 88–98). Therefore, both BRD detection tools used in the present study, CRS and lung ultrasound, are imperfect and may have resulted in the misclassification of calves with BRD.

Our results are most applicable to calves aged 21 to 40 d of age in group housing fed large quantities of milk. We encourage similar studies to determine if the severity of lung consolidation affects feeding behavior. Measuring starter grain intake was not possible in the present study, due to on-farm limitations. However, recent work indicates that time spent at the feed bunk and the number of visits to the feed bunk differs between sick and healthy calves [28], representing an avenue that should be explored further. The minimal changes in feeding behavior around the time of BRD both in the present study and in previous work [9] suggest the need for the validation of other BRD detection tools that can be easily implemented on-farm.

## 5. Conclusions

This was the first study to our knowledge to investigate if calves with subclinical lung consolidation, identified with lung ultrasound and clinical respiratory scoring, exhibit differences in feeding behavior from calves with clinical BRD or calves without BRD. In the present study, which included calves aged 21 to 40 d of age in group housing fed large quantities of milk, feeding behavior did not adequately differentiate between calves with SBRD, CBRD, and NOBRD. As such, dairy producers should be cautious when using automated feeder data as the primary means of detecting calves with BRD.

## Figures and Tables

**Table 1 animals-10-00988-t001:** Proportion (% (n)) of calves in each category of sex (male or female, only male data shown), breed (Holstein or Jersey, only Holstein data shown), and fever (with or without, only ‘with fever’ shown) for each Bovine Respiratory Disease (BRD) status. Calves were enrolled in the study at 21 ± 6 (mean ± SD) d of age and underwent twice weekly health exams. Fever was defined as rectal temperature ≥39 °C.

Variable (n)	BRD Status		
	Clinical BRD (CBRD) (n = 18)	Subclinical BRD (SBRD) (n = 73)	Without BRD (NOBRD) (n = 12)
Sex			
Male	89% (16) ^a^	73% (53) ^a^	83% (10) ^a^
Breed			
Holstein	72% (13) ^a^	84% (61) ^a^	67% (8) ^a^
Fever			
With	100% (18) ^a^	27% (20) ^b^	0 (0) ^c^

^a,b,c^ Different superscripts within rows denote *p* < 0.05.

**Table 2 animals-10-00988-t002:** Raw values for feeding behaviors, by Bovine Respiratory Disease (BRD) status, over the 3 d before, the d of, and the 3 d after BRD detection. Calves were enrolled in the study at 21 ± 6 (mean ± SD) d of age and underwent twice weekly health exams.

Feeding Behavior	BRD Status		
	Clinical BRD (CBRD) (n = 18)	Subclinical BRD (SBRD) (n = 73)	Without BRD (NOBRD) (n = 12)
Average daily drinking speed (mL/min; mean ± SD)	716 ± 230	827 ± 221	879 ± 250
Average daily milk intake (L/d; mean ± SD)	10 ± 2.9	10.6 ± 4.0	10.3 ± 3.4
Average meal size (L/meal; mean ± SD)	1.8 ± 0.9	1.7 ± 0.8	1.6 ± 0.8
Number of rewarded visits (no./d; median; 1st quartile, 3rd quartile)	6 (5, 9)	6 (4, 9)	7 (4, 9)
Number of unrewarded visits (no./d; median; 1st quartile, 3rd quartile)	0 (0, 1)	0 (0, 1)	0 (0, 2)

**Table 3 animals-10-00988-t003:** Multivariable linear regression model for drinking speed (mL/min) collected from an automated milk feeder in 103 preweaned dairy calves. Calves were enrolled in the study at 21 ± 6 (mean ± SD) d of age and underwent twice weekly health exams. The primary predictor of interest was Bovine Respiratory Disease (BRD) status. Drinking speed was analyzed for 3 d prior to, the d of, and 3 days after BRD detection. Estimates for drinking speed and reported *p* are from solutions for fixed effects.

Variable	Drinking Speed Estimate (mL/min)	Standard Error	*p*
Intercept	778.45	58.1	<0.0001
BRD Status			
Clinical BRD (CBRD)	−156.3	65.1	0.02
Subclinical BRD (SBRD)	−61.7	54.9	0.26
Without BRD (NOBRD)	Referent		
Breed			
Holstein	130.9	43.2	0.003
Jersey	Referent

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
