# Peer review of "Automated Feeding Behaviors Associated with Subclinical Respiratory Disease in Preweaned Dairy Calves"

_animals, 2020, doi:10.3390/ani10060988_

Round 1
Reviewer 1 Report
I commend the authors on a thorough and well-presented manuscript. I have minor comments/request for clarifications which I have included in attached draft.

Author Response
Responses to reviewer comments
Cramer et al., Manuscript ID: animals-801762
AU: The authors wish to thank the reviewers for their comments. We feel your comments have improved the manuscript. We have addressed the comments below and in the manuscript. Manuscript changes are highlighted in yellow.
Reviewer 1 (comments obtained from pdf)
I commend the authors on a thorough and well-presented manuscript. I have minor comments/request for clarifications which I have included in attached draft.
AU: Thank you for your comments. We obtained your edits from the pdf and have responded to them below in a line by line manner.
L 28: This abbreviation (CRS) needs to be defined in the abstract.
AU: Changes made (line 27)
Ln 36: Clinical BRD?
AU: Yes, this data is referring to producer-reported clinical BRD. This is clarified in the text (line 37).
Ln 42: unclear - range from 23-69% of all BRD cases in preweaned calves? or 23-69% of preweaned calves suffer from subclinical BRD?
AU: Thank you for this comment. This is referring to prevalence, meaning that 23-69% of calves are identified with subclinical BRD. This has been clarified (line 43).
Ln 56: You might add that they are impractical (if involve restraint and physical examination of individual calves vs. outwardly visible signs, and can't be performed by lay-people in case of ultrasound)
AU: Thank you for this suggestion. This information has been added (line 55).
Ln 57-58: cannot fully understand behavioral changes? or behavioral changes, at least in the context of these described methods, occur once disease is already clinical
AU: This has been clarified in the manuscript (line 58).
Ln 65: again, I might emphasize that this is possible in the research setting but the ultimate goal would be to find an automated, practical on-farm measure
AU: Thank you for this suggestion. This information has been added to the manuscript (line 64-65).
Ln 78: Please add how much starter/other feed available to calves (even if intake not quantified in this study)
AU: Thank you for this information. The farm did not measure the amount of starter. Starter grain was always available to calves free choice. This has been added to the manuscript (line 81).
Ln 96: Define abbreviation here (at first use)- CRS
AU: Changes made. Ln 94.
Ln 134-135: Please specify if calves were kept in the original groups even after diagnosis (not moved to hospital pen) and that all treatments were present in each group
AU: This information has been clarified in the manuscript. Calves remained in their original pen until weaning and were not moved to a hospital pen (Ln 81). All BRD statuses were present in each pen (ln 176-177).
Reviewer 2
Comments and Suggestions for Authors
This paper describes the effects of subclinical BRD on feeding behavior in preweaned calves. Generally, the paper is well-written, however, I have some concerns over the analyses conducted as well as the inclusion criteria used to determine which data was appropriate for these analyses.
Major comments:
1) Why have the authors decided to only analyze the day of the BRD detection? Numerous studies have shown changes in behavior prior to disease onset in calves (Borderas et al., 2009; Knauer et al., 2017; Swartz et al., 2017; Belaid et al., 2019; Belaid et al., 2020) and cows (Huzzey et al., 2007; Itle et al., 2015) (to name a few papers), and these effects can continue well past diagnosis. The authors need to reanalyze their data using a repeated measures analysis. The authors should consider analyzing at least 3 (to 10) days on either side of the date of disease diagnosis. I understand that the premise of this study is simply to see if subclinical BRD alters feeding behavior, but I suspect a farmer is not going to administer treatment to a calf just because the calf's feeding behavior is slightly off for one day - a series of consecutive off-days may merit intervention, but barring any dramatic feeding behavior changes, a single off-day is not sufficient for intervention. Because of this, the current statistical analyses are not logical as application on-farm is very limited. Moreover, considerable effort was done to ultrasound a large number of calves twice weekly for 4 weeks, yet only one day of data was analyzed. Why not present more data?
AU: Thank you for this comment. We have changed the manuscript to include 7 days total: the 3 days prior to, the day of, and the 3 days after BRD diagnosis. This analysis did not change the interpretation of the results, but we feel this approach strengthens the manuscripts. The manuscript has been adjusted accordingly. Please see specifically the methods (lines: 119-120; 149-151; 155-156) and results (lines 169-206). Unfortunately, due to a computer failure in the automated feeding barn during this study, we are unable to analyze more than days -3 to day +3 without severely decreasing our sample size. We hope the 7 day range is acceptable to the reviewer.
2) Handling the calves with diarrhea: The authors make the claim that the study was designed specifically to look at the effects of subclinical BRD on feeding behavior, yet calves with diarrhea are allowed into the analyses. The authors claim that the distribution of diarrhea is not different between BRD statuses, but realistically there aren't enough calves with diarrhea within each BRD status to adequately test for this effect.
I realize some calves will have co-morbidities, but if the study is designed to examine just the effect of subclinical BRD, then you might not want the calves with diarrhea in the analyses. In my opinion, the authors should remove the calves with diarrhea within a certain time frame (something like +/-7 to 10 days of BRD diagnosis). I suppose the authors could attempt to control for the effect of diarrhea in their multivariate analyses by forcing a term in the model. (As a side note, I don't think controlling for fever was ever necessary as calves with BRD are expected to have fever, and the effect of fever on feeding behavior is part of the BRD effect.)
AU: Thank you for the suggestion about removing calves with diarrhea from this data set. This approach did not change the overall interpretation of the results, but we feel this has improved the manuscript. We have updated the entire manuscript accordingly; please specifically see the methods (lines 130-131) and results (168-169).
Minor comments:
Line 52 - needs a citation
AU: Changes made (line 52).
Table 1 - Variable column is very difficult to read.
AU: We have changed the formatting in Table 1 in accordance with the Reviewer’s suggestion below (lines 179-183).
For all tables: Left align the main heading and then put the categories center aligned. So for instance, breed would be left aligned, and then Holstein and Jersey would be center aligned in the table. It should improve the readability of the table. Please do that with every variable.
AU: Thank you for this suggestion. Changes have been made to all tables.
Line 227-248 - In general, I agree with the notion that changes in milk intake is not sufficient to identify calves with BRD; however, other researchers have found a reduction in feed bunk visits from sick veal calves (mostly BRD) (Belaid et al., 2020). Maybe there is a reduction in feed bunk visits involving grain or hay intake, but milk intake/automatic calf feeder visits are generally not affected (prioritization of palatable feed stuffs over less palatable hay/grain?). A discussion on this may be worth mentioning.
AU: Thank you for this suggestion. We have added this information to the discussion (lines 276-278).
Line 277: The future directions do not make a lot of sense to me. Why encourage looking at the effect of subclinical BRD on feeding behavior in younger calves when Knauer et al. (2017) found that clinical BRD has minimal effects on feeding behavior. Can we really expect to see an effect of subclinical BRD when we rarely see an effect of clinical BRD on feeding behavior? Especially when considering that the few changes found in feeding behavior of clinical BRD calves are typically transient only lasting 1 day? Maybe there are better options than feeding behavior?
AU: Thank you for this comment. We have removed this from the discussion and focused on other potential technologies. Lines 276-280
Reviewer 2 Report
This paper describes the effects of subclinical BRD on feeding behavior in preweaned calves. Generally, the paper is well-written, however, I have some concerns over the analyses conducted as well as the inclusion criteria used to determine which data was appropriate for these analyses.
Major comments:
1) Why have the authors decided to only analyze the day of the BRD detection? Numerous studies have shown changes in behavior prior to disease onset in calves (Borderas et al., 2009; Knauer et al., 2017; Swartz et al., 2017; Belaid et al., 2019; Belaid et al., 2020) and cows (Huzzey et al., 2007; Itle et al., 2015) (to name a few papers), and these effects can continue well past diagnosis. The authors need to reanalyze their data using a repeated measures analysis. The authors should consider analyzing at least 3 (to 10) days on either side of the date of disease diagnosis. I understand that the premise of this study is simply to see if subclinical BRD alters feeding behavior, but I suspect a farmer is not going to administer treatment to a calf just because the calf's feeding behavior is slightly off for one day - a series of consecutive off-days may merit intervention, but barring any dramatic feeding behavior changes, a single off-day is not sufficient for intervention. Because of this, the current statistical analyses are not logical as application on-farm is very limited. Moreover, considerable effort was done to ultrasound a large number of calves twice weekly for 4 weeks, yet only one day of data was analyzed. Why not present more data?
2) Handling the calves with diarrhea: The authors make the claim that the study was designed specifically to look at the effects of subclinical BRD on feeding behavior, yet calves with diarrhea are allowed into the analyses. The authors claim that the distribution of diarrhea is not different between BRD statuses, but realistically there aren't enough calves with diarrhea within each BRD status to adequately test for this effect.
I realize some calves will have co-morbidities, but if the study is designed to examine just the effect of subclinical BRD, then you might not want the calves with diarrhea in the analyses. In my opinion, the authors should remove the calves with diarrhea within a certain time frame (something like +/-7 to 10 days of BRD diagnosis). I suppose the authors could attempt to control for the effect of diarrhea in their multivariate analyses by forcing a term in the model. (As a side note, I don't think controlling for fever was ever necessary as calves with BRD are expected to have fever, and the effect of fever on feeding behavior is part of the BRD effect.)
Minor comments:
Line 52 - needs a citation
Table 1 - Variable column is very difficult to read.
For all tables: Left align the main heading and then put the categories center aligned. So for instance, breed would be left aligned, and then Holstein and Jersey would be center aligned in the table. It should improve the readability of the table. Please do that with every variable.
Line 227-248 - In general, I agree with the notion that changes in milk intake is not sufficient to identify calves with BRD; however, other researchers have found a reduction in feed bunk visits from sick veal calves (mostly BRD) (Belaid et al., 2020). Maybe there is a reduction in feed bunk visits involving grain or hay intake, but milk intake/automatic calf feeder visits are generally not affected (prioritization of palatable feed stuffs over less palatable hay/grain?). A discussion on this may be worth mentioning.
Line 277: The future directions do not make a lot of sense to me. Why encourage looking at the effect of subclinical BRD on feeding behavior in younger calves when Knauer et al. (2017) found that clinical BRD has minimal effects on feeding behavior. Can we really expect to see an effect of subclinical BRD when we rarely see an effect of clinical BRD on feeding behavior? Especially when considering that the few changes found in feeding behavior of clinical BRD calves are typically transient only lasting 1 day? Maybe there are better options than feeding behavior?
Author Response

(The authors gave the same response as above.)

Round 2
Reviewer 2 Report
Thank you. Most of my comments have been adequately addressed although there appears to be an error.
My previous suggestion was to remove calves with diarrhea. I suggested this so the authors can conclude with confidence that their findings are indeed a BRD effect, and this effect is not "clouded" by any other disease event (diarrhea). In the previous version, there were 12 control calves without BRD but 2 of these calves had diarrhea. Assuming you removed all calves with diarrhea, you would now have 10 healthy control calves, not 12, however 12 is in both the past and current version of the manuscript. Please check this for accuracy - I don't think you removed the control calves with diarrhea.
The authors also need to force "day" into their model regardless of significance. That is commonly done when conducting repeated-measure analyses. I suspect by removing "day," you subsequently have introduced pseudo-replication into your model, as it is no longer accounting for the repeated measure.
Line 24 - group is misspelled
Author Response
Responses to reviewer comments
Cramer et al., Manuscript ID: animals-801762
Revision 2- minor
Thank you. Most of my comments have been adequately addressed although there appears to be an error.
AU: Thank you for your comments. We have addressed them below.
My previous suggestion was to remove calves with diarrhea. I suggested this so the authors can conclude with confidence that their findings are indeed a BRD effect, and this effect is not "clouded" by any other disease event (diarrhea). In the previous version, there were 12 control calves without BRD but 2 of these calves had diarrhea. Assuming you removed all calves with diarrhea, you would now have 10 healthy control calves, not 12, however 12 is in both the past and current version of the manuscript. Please check this for accuracy - I don't think you removed the control calves with diarrhea.
AU: Thank you for this comment. We looked back at the originally submitted manuscript in response to your comment. The previous manuscript contained an error in Table 1 by stating that 2 NOBRD calves had diarrhea (fecal score ≥ 2). We apologize for this mistake. In response to your comment, we have gone back and verified in this data set (and looked at the raw data to be sure) that there were no calves identified with diarrhea from the NOBRD category during day -3 to day +3.
The authors also need to force "day" into their model regardless of significance. That is commonly done when conducting repeated-measure analyses. I suspect by removing "day," you subsequently have introduced pseudo-replication into your model, as it is no longer accounting for the repeated measure.
AU: Day was included in the model in the repeated statement in PROC MIXED to account for repeated measures, thereby preventing pseudo-replication. We recognize our wording about variables in the final model was confusing. This has been clarified in the statistical section (line 157-158).
Line 24 - group is misspelled
AU: Changes made (line 26).